# Primary and Community Care Transformation in Post-COVID Era: Nationwide General Practitioner Survey

**DOI:** 10.3390/ijerph20021600

**Published:** 2023-01-16

**Authors:** Mònica Solanes-Cabús, Eugeni Paredes, Esther Limón, Josep Basora, Iris Alarcón, Irene Veganzones, Laura Conangla, Núria Casado, Yolanda Ortega, Jordi Mestres, Jordi Acezat, Joan Deniel, Joan Josep Cabré, Daniel Sánchez Ruiz, Marcos Sánchez, Aroa Illa, Ignasi Viñas, Juan José Montero, Francesc Xavier Cantero, Anna Rodriguez, Francisco Martín, Montserrat Baré, Rosa Ripollés, Montse Castellet, Joan Lozano, Antoni Sisó-Almirall

**Affiliations:** 1Family Phisician, Exective Board of the Catalan Society of Family and Community Medicine (CAMFiC), 08009 Barcelona, Spain; 2Primary Care Center Onze de Setembre, Institut Català de la Salut, 25005 Lleida, Spain; 3Primary Care Center Ronda Prim, Mataró, Institut Català de la Salut, 08302 Barcelona, Spain; 4IDIAP Jordi Gol, Institut Català de la Salut, 08007 Barcelona, Spain; 5Primary Care Service Dreta i Muntanya Barcelona, Institut Català de la Salut, 08007 Barcelona, Spain; 6School of Medicine, Vic University, 08500 Barcelona, Spain; 7Primary Care Center Badalona Centre, Institut Català de la Salut, 08911 Barcelona, Spain; 8Primary Care Center Nova Lloreda, Badalona Serveis Assistencials, 08917 Barcelona, Spain; 9Primary Care Center Salou, Institut Català de la Salut, 43005 Tarragona, Spain; 10Primary Care Center Sanllehy, Institut Català de la Salut, 08024 Barcelona, Spain; 11Primary Care Center Casernes, Institut Català de la Salut, 08030 Barcelona, Spain; 12Multiprofessional Teaching Unit of Primary Care in Catalunya Central, Institut Català de la Salut, 08272 Barcelona, Spain; 13Primary Care Center Reus-1, Institut Català de la Salut, 43202 Tarragona, Spain; 14Primary Care Center Sardenya, ACEBA, 08025 Barcelona, Spain; 15Primary Care Center Les Corts, CAPSBE, 08028 Barcelona, Spain; 16Primary Care Center Celrà, Institut Català de la Salut, 17460 Girona, Spain; 17Primary Care Center Montilivi-Vilaroja, Institut Català de la Salut, 17003 Girona, Spain; 18Primary Care Center Rocafonda, Institut Català de la Salut, 08304 Barcelona, Spain; 19Primary Care Center Igualada Urbà, Institut Català de la Salut, 08700 Barcelona, Spain; 20Primary Care Center Santa Eugènia de Berga, Institut Català de la Salut, 08507 Barcelona, Spain; 21Primary Healthcare Research Support Unit, Departament of Primary Care Camp de Tarragona, Institut Català de la Salut, 43202 Tarragona, Spain; 22Primary Care Center Creu Alta, Institut Català de la Salut, 08208 Barcelona, Spain; 23Primary Care Center Temple, Institut Català de la Salut, Terres de l’Ebre, 43500 Tarragona, Spain; 24Primary Care Center Dr. Joan Mirabell, Institut Català de la Salut, 08006 Barcelona, Spain; 25Department of Medicine, University of Barcelona, 08036 Barcelona, Spain

**Keywords:** primary healthcare, health policy, work organization, COVID-19

## Abstract

Introduction: The health emergency caused by COVID-19 has led to substantial changes in the usual working system of primary healthcare centers and in relations with users. The Catalan Society of Family and Community Medicine designed a survey that aimed to collect the opinions and facilitate the participation of its partners on what the future work model of general practitioners (GPs) should look like post-COVID-19. Methodology: Online survey of Family and Community Medicine members consisting of filiation data, 22 Likert-type multiple-choice questions grouped in five thematic axes, and a free text question. Results: The number of respondents to the questionnaire was 1051 (22.6% of all members): 83.2% said they spent excessive time on bureaucratic tasks; 91.8% were against call center systems; 66% believed that home care is the responsibility of every family doctor; 77.5% supported continuity of care as a fundamental value of patient-centered care; and >90% defended the contracting of complementary tests and first hospital visits from primary healthcare (PHC). Conclusions: The survey responses describe a strong consensus on the identity and competencies of the GP and on the needs of and the threats to the PHC system. The demand for an increase in health resources, greater professional leadership, elimination of bureaucracy, an increase in the number of health professionals, and greater management autonomy, are the axes towards which a new era in PHC should be directed.

## 1. Introduction

The Spanish healthcare system is universal and financed through taxes, so patients do not make a direct payment for the service received. It is basically organized through a network of nearby primary care centers (PCCs) through a strategic geographic distribution of continuing care points/emergency points and hospitals. General practitioners, pediatricians, nurses, dentists, midwives, and social workers work in primary care centers that are characterized by being accessible to the population. Hospital visits are requested by GPs or pediatricians if the visits are necessary to resolve the patient’s clinical process [1]. This concentration of services in primary care, before COVID-19, involved a consistently high-volume population because assistance was basically face-to-face.

The global emergency caused by the COVID-19 pandemic has put the entire health system under great strain. Primary healthcare (PHC) teams had to make sudden, substantial changes to the usual way of relating to users [2], including fewer in-person visits and more telemedicine appointments [3]. The need to reduce the risk of contagion led to the demassification of health centers, with first appointments being telephonic or virtual, after which, physicians determined the necessary resources: face-to-face appointments, home visits, or resolution through telemedicine. This new modality of non-face-to-face care has occurred in Spain and in many other countries [4,5] and has generated much debate among professionals.

The described characteristics of our healthcare system justify the importance of a well-funded primary care system, since it guarantees equitable access to the health system and often resolves health problems; when these services are not possible, the primary care system refers to hospital consultation. Declarations of Alma-Ata and Astana said that primary healthcare is essential healthcare based on practical, scientifically sound methodology and technology made universally accessible to individuals and families in the community, with a cost that the community and country can afford to maintain at every stage of development [6,7]. In this respect, a Norwegian study, published in 2022, showed that the length of regular general practitioner–patient relationships were significantly associated with lower use of after-hours services (up to 30%), fewer acute hospital admissions (up to 28%), and lower mortality (up to 25%), with a causal association [8]. However, in Spain, PHC faced this health crisis with important issues still unresolved since the economic crisis of 2008: an obvious budget shortfall with a shortage of staff, and many health centers pending renovations or expansion with serious problems due to lack of space and ventilation. Reports from Amnesty International and the Federation of Associations in Defense of Public Health (FADSP) coincided in underlining the reduction in the percentage of expenditure dedicated to PHC in proportion to total public health expenditure in 2010–2018, which remained between 11.48% (Madrid), 12.98% (Catalonia), and 17.45% (Andalusia) in 2018 (public data from the Ministry of Health), far from the 25% recommended by the World Health Organization (WHO) [9,10].

Some authors have represented the growing difference between hospital health expenditure and primary care expenditure as the “snake’s mouth”, in clear allusion to the gap between the uneven growth curves of the two areas [11]. The situation in PHC fits into a context of low total health expenditure, far removed from that of European reference countries [12].

Despite this, Catsalut (insurer of the health provision of the Department of Health of Catalonia) records show that total PHC activity in 2020, when the pandemic started, increased by 7.2% (52,898,350 visits) compared with 2019, with a reduction in face-to-face visits of 42.2% (22,876,552 visits), an increase in home care of 7.9% (1,916,730 home visits), an increase in telephone visits of 467.8% (18,618,595 visits), and an increase in information technology care of 101.6% (9,486,478 virtual visits) [13].

Given the budgetary constraints in primary care and the changes in care generated in order to adapt to the pandemic, we proposed to determine the opinions of members of the Catalan Society of Family and Community Medicine (CAMFiC) on different aspects related to our specialty, on the working model of general practitioners and, by extension, of PHC teams, and on their assessment of the current situation and their projection of future prospects.

## 2. Methodology

We made a cross-sectional descriptive study based on an ad hoc survey administered by sending three successive emails between 1 February 2021 and 22 March 2021. At that time, the health system was in a phase of decreasing cases of a fourth COVID-19 wave. The survey was sent to GPs who are members of CAMFiC, which, with 4645 members, is the largest scientific society in Catalonia. The responses obtained were voluntary and anonymous. The questionnaire was developed and agreed upon by the Governing Board of the Catalan Society of Family and Community Medicine, composed of 25 family physicians representative of all the counties of Catalonia.

The questionnaire consisted of three sections [14]. The first section contained data on filiation: age, sex, territory (health regions), work environment (urban, semi-urban, rural, emergencies), and years of professional experience. The second section contained 22 multiple-choice questions with a Likert-type rating scale from 1 to 5 (1: completely disagree, 2: partially disagree, 3: neutral, 4: agree, and 5: completely agree). Figure 1 shows, in a visual way, the results of all the questions with the 5 response options according to the Likert scale. In the results portion of the second section, the results were grouped with the values 1 and 2 = disagree, and 4 and 5 = agree in order to facilitate the reading and the interpretation of the tendency of the answers. Moreover, to clarify the trend of the answers and their interpretation, the answers were grouped with values 1 and 2 = disagree, and 4 and 5 = agree. The 22 questions examined 5 dimensions: (1) feelings during the pandemic: transmission of strength by the team, anguish/insomnia, lack of organization, professional integrity despite adversity, and time lost to bureaucracy; (2) work agendas: appointment system, design of the medical agenda, and ratio of face-to-face to non-face-to-face visits; (3) general practitioners (GPs) and PHC centers: role of the GP, home care, and end-of-life care; (4) PHC centers and their organization and coordination; and (5) PHC and the health system: 10-year projections, evolution of the health system, telemedicine, purchase of hospital services and intermediate products from PHC, PHC payment systems, integration of residential equipment, decentralization of procedures and hospital budgets transferred to PHC, management autonomy, and management of PHC centers by social enterprises. The third section contained an open unlimited question: “Would you like to contribute any thoughts? State what you think the working model of GPs in the post-COVID era should look like”.

### 2.1. Statistical Analysis

Categorical variables were expressed as absolute frequencies and percentages (%) and continuous variables as means and standard deviation (SD). The answers to the open question were analyzed by two independent observers for the degree of coherence and relevance, without excluding any answer. The statistical analysis was performed using R version 3.6.1. (R Foundation for Statistical Computing, Vienna, Austria) for Windows.

### 2.2. Ethical Aspects

This is an observational and quantitative study that does not involve any type of intervention on people; no exploration is carried out, nor is any additional information obtained, but rather, the value of the variables is extracted strictly from the information given voluntarily by respondents. The procedures followed Spanish and Catalan laws. Researchers followed the ethical standards of the Declaration of Helsinki for biomedical studies and the activities described followed the Code of Good Practice in clinical research.

## 3. Results

Responses were obtained from 1051 general practitioners (response rate 22.6%), of whom 773 were female (73.54%). The demographic characteristics of participants are shown in Table 1.

In the first block of questions, on self-perception during the pandemic, 83.2% of participants stated they spent too much time on bureaucratic tasks, mainly related to work disabilities. The second section of the questionnaire consisted of 22 questions distributed in 5 blocks. It can be consulted in Table A1 of Appendix A.

In the second block, on work schedules, 91.8% were opposed to the call center system (centralization of calls in units external to the team) to make patients’ appointments. Likewise, 97% and 89% believe that neither the Ministry of Health nor the General Directorate of Primary Healthcare, respectively, should design the agendas of GPs, while 84% believed that each doctor should do so in accordance with the PHC center management.

In the third block, on the GP and the team, 66% stated that home care was the responsibility of each GP and, by extension, the PHC centers. In the case of end-of-life care, 63% of respondents defended the same model.

In the fourth block, on PHC centers and their composition, 77.5% supported continuity of care, understood as ensuring patients see the same doctor and nurse, 75% and 73%, respectively, agreed with the incorporation of psychologists and physiotherapists to PHC centers, although almost 70% believed that the basic structure should be reinforced first.

In the fifth block, on PHC and the health system, looking towards the future, more than 90% believed that mechanisms should be established that allow the contracting by PHC of complementary tests and first hospital visits, 79.8% believed it necessary to promote telemedicine and telemonitoring, 78% believed that residential health should be integrated with PHC centers, and 85% agreed with decentralizing hospital procedures and budgets to PHC centers.

The results from Section 2 are shown in Figure 1.

Finally, 52.5% (552) of participants expressed their opinion in the free text question. The thematic grouping was categorized into the following results: increased resources for PHC, better PHC leadership, elimination of bureaucracy, intensification of information and communication technologies (ICT), more doctors and other professionals, more time for clinical care and less bureaucracy, and greater autonomy in the management of one’s own work.

## 4. Discussion

This study of a large sample of general practitioners surveyed after the fourth wave of the COVID-19 pandemic shows that excessive bureaucracy is one of the most important perceived problems in PHC, with a negative impact on the self-perception of work quality. Likewise, participants stated that control of the care agenda should belong to GPs, and home care and end-of-life care are core, essential PHC spaces. Our study also shows that GPs identify continuity of care as a raison d’être of PHC. GPs also emphasized the need to contract hospital services (diverting a part of the hospital budget to PHC for management), the necessary promotion of telemedicine, and attention to residential care homes for the elderly as PHC spaces.

GPs were busy and worried about caring for patients with COVID-19 and other illnesses and were flooded by a huge bureaucratic demand. This is reflected by 83.2% of respondents, who state they felt that they were wasting time on patients’ sick leave paperwork and other bureaucratic tasks. Studies have established a clear relationship between burnout and the bureaucratic burden, which reduces the time left for clinical activity [15,16]. The prioritization of COVID-19 care, both clinically and administratively, caused a displacement of care with respect to chronic diseases and the diagnosis of new health problems, the usual tasks of PHC in Spain and in other countries [17,18,19,20].

In relation to the demand for autonomy in matters related to the organization of care, 91.8% were opposed to the call center system, with consensus on the need for triage for each GP or the support of a specific administrative worker. Likewise, 97% and 89% believed that neither the Department of Health nor the Primary Healthcare Directorate, respectively, should design GP work agendas, while 84% believed that each doctor should do so in accordance with the team management. We believe that the high level of consensus observed in the questions about care is related to knowledge acquired during the pandemic on the management of consultations and of the health teams [21]. We face the challenge of avoiding face-to-face consults when they are not necessary, in order to prioritize adding value when they are, thereby maintaining the confidence of the population [22].

Nearly 70% of respondents believed that home care and end-of-life care are the responsibility of individual GPs and their teams. More than 80% disagreed with the management of home care and end-of-life care by external units. Home care is the activity that best represents the essential attributes of primary care (continuity of care, integrality, coordination) and which suffers the most when these attributes do not permeate PHC activity or when budget cuts are applied. This type of care should be seen as a “normal” activity of PHC centers. However, this normality should not hide the huge range of characteristics this work has. Greater accessibility and intensity of care, etc., must be reflected and measured so that GPs can give adequate time to do so, as the patients and their families deserve [23,24].

Regarding the organization of professionals, 77% chose continuity of care as a central value to be preserved. Several studies have emphasized the demonstrated benefits of continuity of care in such important issues as reduced mortality and hospitalizations and reductions in the use of emergency services [8,25,26,27]. In our study, 65% of participants agreed with adding a health administrator to the reference teams of doctors and nurses. Regarding the composition of PHC centers, more than 70% of participants agreed with the incorporation of psychologists and physiotherapists. There are similar experiences in other countries [28,29]. As important as the profile of these professionals was, the formula with which this incorporation into PHC centers would be carried out to guarantee their integration was also important. However, almost 70% believed that the basic PHC structure (GPs and nurses) should first be reinforced.

Of the different options on the evolution of the health system, more than 90% of participants agreed with establishing mechanisms to enable the contracting by PHC of complementary tests and first hospital visits, and 85% considered that, in terms of cost-effectiveness, hospital procedures and budgets should be decentralized to PHC centers, an observation reinforced by other studies [30,31].

The promotion of telemedicine and telemonitoring was thought necessary by 80% of participants. The COVID-19 pandemic was a significant accelerator for telemedicine, since most visits were made in its different modalities. There was exponential growth of online consultations, which were effective and avoided face-to-face consultations in 55–79% of cases [3,32]. The mass implementation of telemedicine will require improvements in computer resources and training in their use from university studies onward, as well as mitigation of the possible inequities that may arise [33,34]. Its acceptance by users and providers, and its advantages in terms of cost-effectiveness will likely make telemedicine the form of consultation [35].

Almost 80% of participants believed that the health teams of residential centers should be integrated with PHC centers and accommodated with the necessary resources for efficient care, and 65% agreed on creating units coordinated and managed by PHC for the care of people with complex needs. This could be a new system of organization of health services based on the versatility of general practitioners and their natural proximity to the population. Several studies have promoted the streamlining of healthcare by adapting devices and care systems according to different profiles of patients and pathologies, thereby avoiding reactive care to demand [36,37].

The study has some limitations. Although a survey in which more than a thousand GPs participated might seem highly representative, this represents just over 20% of the total number of GPs in Catalonia. We observed a great deal of thematic variability in the free text section of the survey. We believe that this section deserves a different sub-analysis using a qualitative methodology, which will be the subject of a different future study. Another limitation is that the questionnaire used was specifically prepared for this study, and therefore cannot be contrasted with previous results or with PHC doctors from other locations. Likewise, it was not possible to make comparisons with other PHC systems given the characteristics of this level of care and the health systems of each country.

## 5. Conclusions

In conclusion, after analyzing the questionnaire, a strong consensus was observed in many of the questions posed, especially in those referring to the activity of general practitioners demanding to reduce bureaucracy, to maintain continuity of care, and demanding autonomy in the management of centers and care. There was also strong agreement on how the health system and PHC should evolve. In this sense, there was support for measures that promote a greater effective relevance of PHC within the health system, such as purchasing power, for example, and a greater allocation of resources.

## Figures and Tables

**Figure 1 ijerph-20-01600-f001:**
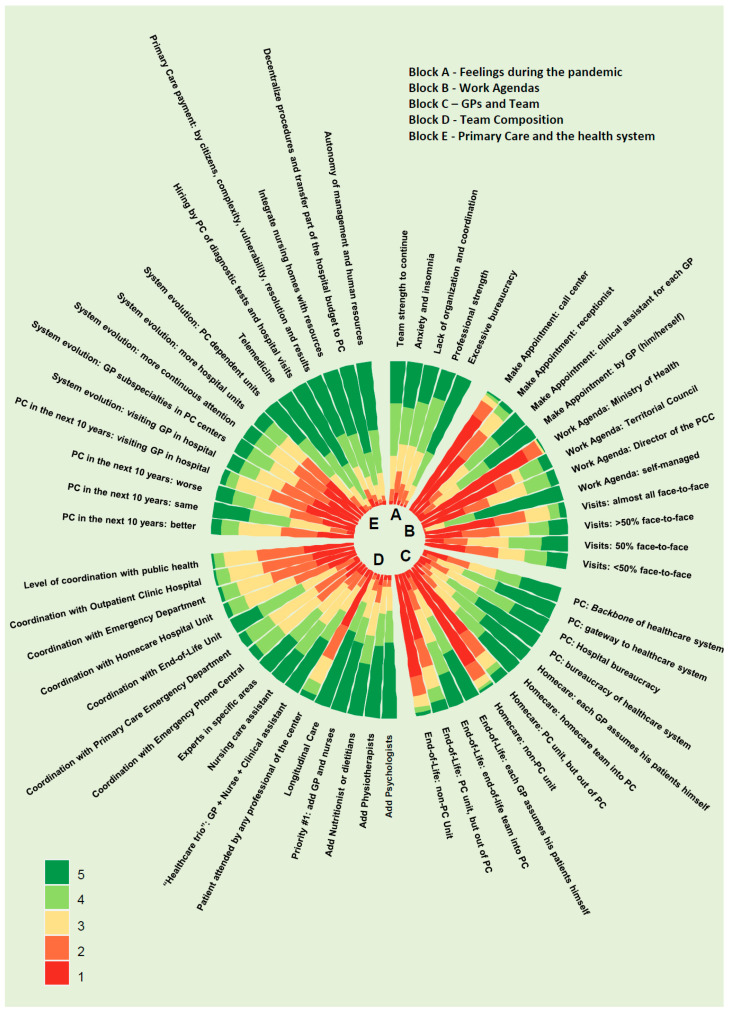
Second section of the questionnaire with 22 questions distributed in 5 blocks.

**Table 1 ijerph-20-01600-t001:** Baseline features.

Sex	Female	773 (73.54%)
Age	25–35	126 (11.98%)
	36–45	268 (25.49%)
	46–55	409 (38.91%)
	>55	248 (23.59%)
Work experience (in years)	<5	99 (09.41%)
	6–15	238 (22.64%)
	16–25	396 (37.67%)
	>25	318 (30.25%)
Area of work	Urban	698 (66.41%)
	Semi-urban	187 (17.79%)
	Rural	155 (17.74%)
	Emergency	11 (01.04%)
Health Region	Barcelona	57%
	Lleida	13%
	Central Catalonia	10%
	Tarragona	9%
	Girona	8%
	Teres de l’Ebre	2%
	Alt Pirineu i Aran	1%

## Data Availability

The full dataset is available from the corresponding author upon reasonable request.

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
