# Peer review of "Primary and Community Care Transformation in Post-COVID Era: Nationwide General Practitioner Survey"

_ijerph, 2023, doi:10.3390/ijerph20021600_

Round 1

Reviewer 1 Report

The manuscript presents the results of a cross-sectionnal survey administered to family doctors who are members of the Catalan Society of Family 78 and CAMFiC (Community Medicine). Thank you for the opportunity to review this manuscript. I provide some major and minor comments that I hope will help to improve the reporting of your work.

Major comments :

1/ In the introduction section, you started by the COVID-19 crisis and the need to demassify of health centers, then you emphasize the decrease of expenditure dedicated to primary healthcare which began since the 2008 crisis. It is difficult to follow the sequence of ideas, and to correlate precisely the mechanics of these events.

2/ The initial hypothesis and the purpose of the study are not clearly introduced.

3/ The material paragraph is missing in explanation of the development of the questionnaire. Was it developed by a working group of family physicians, emergency physicians, epidemiologists, methodologists or policy makers? Also, can you clarify if there are other questionnaires of this type in previous studies in the scientific literature?

4/ How do you explain the initial choice of a Likert-type rating scale from 1 to 5 when you explained that the final answers have been grouped on the two extreme values ? Does this mean that the choice of 5 response modalities was not the right one? Yet the Figure 1 shows the 5 response modalities.

5/ Quality of work and mental health issues are not included in the questionnaire. Don't you think this is also an important aspect that can capture the practice and conditions in which family doctors practice, especially since there are already standardized measurement scales?

6/ What was the minimum sample size of respondents to draw meaningful conclusions?

7/ Knowing that more than half of the respondents brought free text, the exploitation of the verbatims is insufficient and does not make it possible to account for a part of your objective which is the prospects of future evolutions of the system.

8/ The response rate is correct compared to other surveys in the scientific literature. However, the results would be more valuable if the representativeness of the territorial areas was investigated.

9/ About ethical authorizations ?

Minor comments :

a/  In the abstract, please modify the term “COVID” in “COVID-19”, “careas” in “care as”.

b/ In the results, please fix the term “were are opposed”.

c/ The resolution of the Figure 1 needs to be improved.

Reviewer 2 Report

The introduction could be extended with the inclusion of some key aspects such as:

a) the importance of the strong PHC in the defense to maintain the safety and health of people especially in low and middle income countries where people have limited access to hospitals and specialized care. Addressing the challenges of providing basic and essential services in response to the epidemic disease and its recovery, needs strong and resilient health systems, especially at the PHC level (ref. Rezapour et al. The impact of the Covid-19 pandemic on primary health care utilization: an experience from IranBMC Health Services Research (2022) 22:404.https://doi.org/10.1186/s12913-022-07753-5);

b) the description of how the PHC system is structured in Spain;

c) the importance of investments in PHC to achieve universal health coverage (UHC);

d) references to some policy documents that refer to the PHC issue (e.g. The Alma Ata Declaration, 1978);

e) a brief description of how PHC has changed over the centuries;

These are just some points of reflection and expansion of the introduction.

The bibliographic references seem too focused on the Spanish case. I think that at least in the introduction it is necessary to refer to the global context and therefore to documents of international significance (e.g. WHO, UN, ILO, etc).

Please review the text and replace “indifferent” as the central value of the Likert scale with the term “neutral”. To indicate the general practitioner, use the term general practitioner

Round 2

Reviewer 1 Report

x

Reviewer 2 Report

This version includes all the reviewers' comment and suggestions. It can be published in present form